# LoQT: Low-Rank Adapters for Quantized Pretraining

**Sebastian Loeschcke***
University of Copenhagen
sbl@di.ku.dk

**Mads Toftrup***
Aarhus University
toftrup@cs.au.dk

**Michael J. Kastoryano**
University of Copenhagen
mika@di.ku.dk

**Serge Belongie**
University of Copenhagen
s.belongie@di.ku.dk

**Vésteinn Snæbjarnarson**
University of Copenhagen
vesn@di.ku.dk

## Abstract

Despite advances using low-rank adapters and quantization, pretraining of large models on consumer hardware has not been possible without model sharding, offloading during training, or per-layer gradient updates. To address these limitations, we propose Low-Rank Adapters for Quantized Training (LoQT), a method for efficiently training quantized models. LoQT uses gradient-based tensor factorization to initialize low-rank trainable weight matrices that are periodically merged into quantized full-rank weight matrices. Our approach is suitable for both pretraining and fine-tuning models. We demonstrate this for language modeling and downstream task adaptation, finding that LoQT enables efficient training of models up to 7B parameters on a 24GB GPU. We also demonstrate the feasibility of training a 13B model using per-layer gradient updates on the same hardware.

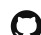 https://github.com/sebulo/LoQT

## 1 Introduction

Training large neural networks requires substantial hardware and energy resources. Reducing these requirements is important for both cost efficiency and environmental reasons, while also lowering the entry barrier for researchers and practitioners in general. In this work, we target the memory component—a key part of the hardware requirements. Memory use during training comes primarily from storing the weights of the model, the optimizer states, and activations. To target the memory footprint of the weights, various applications of quantization [1, 2, 3, 4] have been used. For targeting the optimizer states,

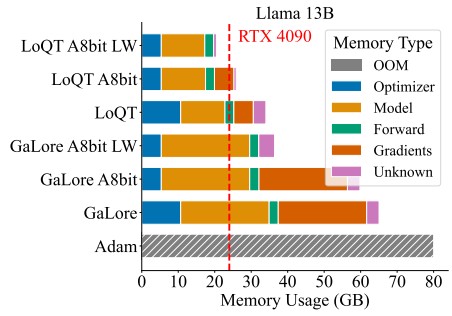

Figure 1: Memory usage of Llama 13B, rank 1024. LW: per-layer gradient updates. A8bit: Adam 8bit.

variations on low-rank adaptation (LoRA) [5, 6, 3, 7] have been suggested to decrease the number of trainable parameters for fine-tuning, in combination with the use of low precision representations. Low-rank approaches for projecting gradients to a lower rank have also been suggested [8]. In this work, we combine these approaches to address the model size and optimizer states, resulting in a highly memory-efficient configuration that is also suitable for pretraining.

---

*Equal contribution.

38th Conference on Neural Information Processing Systems (NeurIPS 2024).

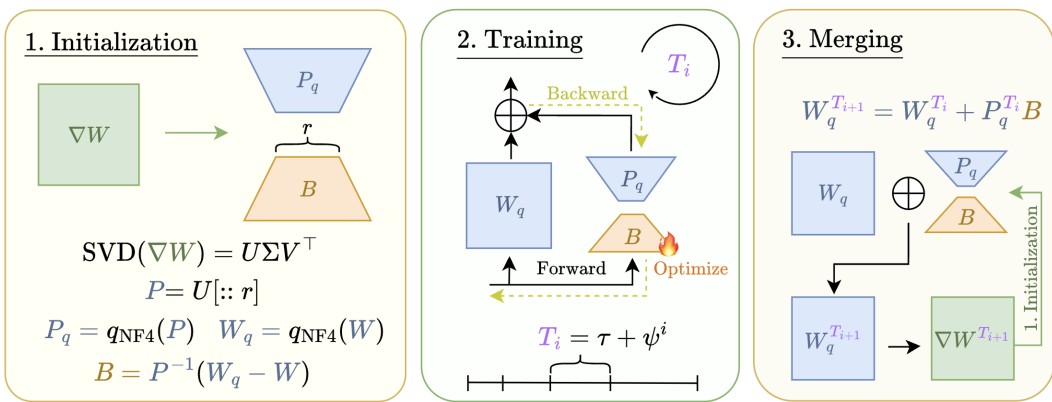

Figure 2: Overview of LoQT. (1) Low-rank factors $P$ and $B$ are periodically initialized from the gradient of the dequantized model weights $\nabla W$, (2) then only $B$ is trained while $P_q$ and $W_q$ are kept quantized and frozen, over an exponentially increasing interval until $T_i$, (3) the low-rank factors are merged back into the quantized model. The process is repeated until training halts.

In typical training configurations, the optimizer states often take up more space than the model itself, as methods such as Adam [9] keep track of two parameters for each parameter of the model. While LoRA is memory efficient for parameter-efficient fine-tuning of pretrained models, it has not been shown to work as a pretraining method by itself [7]. GaLore [8] significantly reduces the memory needed for the optimizer parameters by storing the optimizer state in a low-rank projection, which is then projected up when applied to the model weights. Combining this method with quantization would further shrink the footprint of the model but this is not straightforward. Updating the weights of a highly quantized model directly in low-precision space has not been shown to work. This is mainly due to the higher-precision gradient updates having too small of an impact on the lower-precision quantized states.

To address these shortcomings, we propose *Low-Rank Adapters for Quantized Training* (LoQT). LoQT initializes two low-rank factors, $P$ and $B$, for each weight matrix $W$. $P$ is initialized using a projection of $W$'s gradients into a low-rank subspace, and $B$ is initialized to minimize the quantization error. In our method, $B$ is the only matrix being actively optimized. Only optimizing $B$ means that the size of the gradients and optimizer state shrinks significantly compared to full training or LoRA. The product $PB$ is periodically merged into the full rank matrix $W$ with exponentially increasing gaps to account for smaller updates as the model converges, ensuring we accumulate large enough updates. As $W$ and $P$ do not receive gradient updates, they can be kept quantized, optimizing memory usage even further. It is the large accumulated updates that make it possible to update a quantized model—as the addition of smaller changes would not register in the quantized state. A high-level overview of our approach is given in Fig. 2.

We show that LoQT works well both with and without quantization, enabling not only a lower memory footprint in the optimizer state but also over the model parameters. Our results show that we get competitive performance to prior methods using significantly less memory, in particular when quantizing the model weights in an application such as training a large language model (LLM). We also demonstrate comparable performance in language adaption, which we demonstrate on a curated Icelandic text dataset [10]. Finally, we show that LoQT also works for fine-tuning pretrained models on down-stream tasks, by training and evaluating on the GLUE [11] benchmark for natural language understanding and the GSM8K [12] dataset for mathematical reasoning. We ablate several properties of the suggested approach, demonstrating the importance of each component of LoQT. For instance, we find that an exponentially increasing projection gap is particularly crucial for the training of quantized models. An overview of memory savings is given in Fig. 1. We find that LoQT enables efficient training of 7B models on consumer-grade hardware with 24GB of memory, and makes it feasible to train models with up to 13 billion parameters without model parallelization, by making use of per-layer gradient updates [13].

# 2 Efficient Pretraining With LoQT

We now briefly introduce how LoQT works by initializing and training low-rank adapters. The adapters are initialized by taking the singular value decomposition (SVD) of a given layer's gradients. We use $W$ to indicate the full weight matrix of a given layer and $P$ for the left factor constructed from the SVD decomposition of the gradient matrix, $\nabla W = U\Sigma V^\top$, such that $P$ consists of the first $r$ columns of $U$—corresponding to the singular vectors with the $r$ largest singular values of $W$, where $r$ is a given target rank. The update rule for a timestep $T_i$ is then given by $W_{T_i} = W_{T_{i-1}} + PB$. For the steps between $T_i$ and $T_{i+1}$ only the weights of $B$ are updated, while $P$ and $W_{T_{i-1}}$ remain constant. We describe this in more detail below, followed by a discussion on periodic updating of the factor $P$, enabling of quantized pretraining, error compensation, and exponential update intervals. Pseudo-code for LoQT is shown in Fig. 3.

## 2.1 Background: GaLore

Zhao et al. [8] find that gradients exhibit a low-rank structure during training. They exploit this insight by projecting the gradient to a low-rank subspace and applying the Adam optimizer before projecting back to the original dimensions. By doing this, the memory-intensive optimizer states required by Adam are shrunk significantly for low enough ranks.

**Definition 2.1** (Gradient Low-rank Projection, def. 3.4 in [8])**.** Gradient low-rank projection (**GaLore**) denotes the following gradient update rules, where $\eta$ is the learning rate, $\rho$ is the Adam optimizer, $W \in R^{m \times n}$ is the weight matrix being trained, and $T$ represents the total number of training iterations until the recomputation of the projection matrix:

$$W_T = W_0 + \eta \sum_{t=0}^{T-1} \tilde{G}_t, \text{ where } \quad \tilde{G}_t = P_t \rho_t(P_t^\top G_t Q_t) Q_t^\top, \tag{1}$$

where $r$ is a given target rank and $P_t \in R^{m \times r}$ and $Q_t \in R^{n \times r}$ are the top-$r$ singular vectors from the SVD decomposition of the gradient matrix at each iteration $t$. In practice, this can be approximated by only applying a one-sided projection, as in

$$W_T' = W_0 + \eta \sum_{t=0}^{T-1} P_t \rho_t(P_t^\top G_t) \text{ or } W_T' = W_0 + \eta \sum_{t=0}^{T-1} \rho_t(G_t Q_t) Q_t^\top. \tag{2}$$

Additionally, Zhao et al. [8] empirically show that it is sufficient to keep the projection matrix fixed and only update it once every $T$ iteration.

## 2.2 Low-rank Gradients as Adapters

We now describe how we initialize the parameters we optimize with LoQT. We start with the GaLore formulation from above and adopt the memory-performance trade-off of using only a one-sided projection (eq. 2), we compute $P^\top G$ if $m \leq n$ and $GQ$ otherwise. Our goal is to separate trainable weights and static weights, which we achieve by rewriting GaLore in terms of low-rank adaptors. We assume that $m \leq n$, if $m > n$ the same reasoning holds for $Q_t^\top$. Using the fact that $P_t$ is fixed on the interval $[0, T]$ we get

$$W_T = W_0 + \eta \sum_{t=0}^{T-1} P \rho_t(P^\top G_t) \tag{3}$$

$$= W_0 + \eta \underbrace{P}_{\in \mathbb{R}^{m \times r}} \underbrace{\sum_{t=0}^{T-1} \rho(P^\top G_t)}_{B \in \mathbb{R}^{r \times n}}. \tag{4}$$

It is clear from (4) that we can keep track of low-rank updates using rank-$r$ adaptors. We note that in the interval $[0, T]$ only $B$ is updated, creating the desired separation. If implemented directly, we would need to compute the gradient with respect to $W$ and then project it down using $P^\top G_t$. We find that this step is unnecessary; it is sufficient to train $B$ using standard gradient descent.

**Equivalence of Gradient Updates** We point out that optimizing the low-rank matrix $B$ via gradient descent is equivalent to the projected gradient updates on $W_t$ described in Definition 2.1. Let $G^W = \frac{\partial \mathcal{L}}{\partial W}$ and $G^B = \frac{\partial \mathcal{L}}{\partial B}$ denote the loss gradients with respect to $W$ and $B$, respectively. Consider the forward pass $y = xW + xPB$, where $W$ is the weight matrix, $P$ is the projection matrix, and $B$ is the low-rank update matrix. By the chain rule:

$$G^B = (xP)^\top \frac{\partial \mathcal{L}}{\partial y} \tag{5}$$

$$= P^\top x^\top \frac{\partial \mathcal{L}}{\partial y} \tag{6}$$

$$= P^\top G^W \tag{7}$$

This derivation establishes that computing gradients with respect to $B$ is equivalent to projecting the gradients with respect to $W$ onto the low-rank subspace defined by $P$. Therefore, GaLore's low-rank gradient updates are identical to those obtained through backpropagation in LoRA.

## 2.3 Pretraining with LoRA

Previous work [5] has shown that training low-rank weight matrices works well for fine-tuning pretrained weights. However, it has been shown that starting with randomly initialized weights, training low-rank factors, and periodically merging them into a frozen weight matrix $W$, does not work when starting with a randomly initialized matrix [7]. We now address this to enable full training using low-rank weight matrices.

Inspired by prior work [7, 8], we periodically update a given layer $W_{T+1} = W_T + P_T B_T$ at fixed steps $T \in \mathcal{T}$. This approach allows $W$ to evolve as a sum of low-rank matrices aligning with GaLore's strategy of updating the gradient subspace during training:

$$W_t = W_0 + \Delta W_{T_1} + \Delta W_{T_2} + \ldots + \Delta W_{T_n}, \tag{8}$$

where $t = \sum_{i=1}^{|\mathcal{T}|} T_i$ and $\Delta W_{T_i} = P_{T_i} B_{T_i}$ represents the product of the learned matrix $B$ over the interval $T_i - T_{i-1}$ modulated by the gradient projection matrix $P_{T_i}$. After each periodic update at iterations $T_i \in \mathcal{T}$, we reinitialize the low-rank factors $P_T$ and $B_T$. As in [8], we compute the gradient of $W_T$ over a single batch, focusing only on $\nabla W_T$ without storing optimizer states for it, reducing the memory compared to full-rank training.

For each updated $W_t$ and reinitialized $P_t$ and $B_t$, a new gradient subspace is established for exploring the next $T_{i+1} - T_i$ steps. Our method treats $W_t$ as the full-rank repository of accumulated updates. Although it is periodically updated, $W_t$ is not part of the optimizer state computations, and the gradients during the single forward pass are offloaded to CPU/RAM. Since the SVD calculations are done layerwise, only the current layer needs to be on GPU, or the SVD can be calculated on CPU. $P_t$ defines the general gradient subspace and trajectory for the upcoming $T_{i+1} - T_i$ steps, and $B_t$ is adjusted to navigate within the direction set by $P_t$. As only $B_t$ is trained, the number of parameters requiring optimizer states is drastically reduced.

## 2.4 Quantized Training

Recall the update rule of our model, $W_{T_i} = W_{T_{i-1}} + PB$, given that $B$ is the only matrix accumulating gradients and undergoing changes, the other matrices $W$ and $P$ can be kept quantized. This approach allows storing the weights in NF4 precision [3] (see §5.1 for a detailed account) without requiring high-precision gradients and weights to update $W$ and $P$. To the best of our knowledge, we are the first to enable efficient 4-bit quantized pretraining using gradient descent without storing the weights in 16-bit precision.

We quantize the weights $q_{\text{NF4}}(W) = W_q$ and $q_{\text{NF4}}(P) = P_q$ as described in §5.1. During the periodic updates at interval time steps $(\sum_{i=1}^{n} T_i)_{n=1}^{\max}$, $P_q$ and $W_q$ are dequantized using the inverse function, $P_{\text{BF16}} = q_{\text{NF4}}^{-1}(P_{\text{NF4}})$ and $W_{BF16} = q_{\text{NF4}}^{-1}(W_{\text{NF4}})$. After this, $W_{T_i} = W_{T_{i-1}} + P_{T_{i-1}} B_{T_{i-1}}$ is computed and quantized. The quantization and dequantization processes are applied layer by layer, ensuring that not all layers are simultaneously in a non-quantized state to reduce memory usage. Moreover, the quantization state itself is re-quantized for further efficiency following [3]. We implement LoQT using weight-only quantization, this means that the quantized weights are loaded into memory and then dequantized before computing the matrix multiplications.

**Algorithm 1** LoQT: Low Rank Adapters for Quantized Training

---

**Require:** $W$: Weight, $T$: Update steps, $\eta$: LR, $r$: rank, $q_N(\cdot)$: N-bit quantization function.
1: $G_W \leftarrow \nabla_W \mathcal{L}(W)$
2: $W_Q, P_Q, B \leftarrow \text{Initialize}(W, G_W)$
3: **for** each $t$ in training steps **do**
4:    **if** $t \in T$ **then**
5:       $W \leftarrow W_Q + s \cdot P_Q \cdot B_t$
6:       $G^W \leftarrow \nabla_W \mathcal{L}(W)$
7:       $W_Q, P_Q, B_t \leftarrow \text{Initialize}(W, G^W)$
8:    **else**
9:       $B_{t+1} \leftarrow B_t - \rho(G_t^B)$
10: **return** $\theta$

---

**Algorithm 2** Initialization Procedure

---

1: **Initialize**$(W, G^W)$:
2: $U, S, V^T \leftarrow \text{SVD}(G^W)$
3: $P \leftarrow U[:, :r]$ {First $r$ singular vectors}
4: $P_q \leftarrow q_N(P)$
5: $B \leftarrow 0$
6: $\hat{W} \leftarrow W$
7: **for** each $c$ in compensation steps $C$ **do**
8:    $Q_c \leftarrow q_N(\hat{W})$
9:    $B \leftarrow P^+(\hat{W} - Q_c)$
10:    $\hat{W} \leftarrow W - PB$
11: **return** $Q_c, B, P_q$

---

Figure 3: Pseudo-code for LoQT.

## 2.5 Compensating for Quantization Errors

As the quantization process inevitably results in rounding errors there is a discrepancy between the non-quantized and quantized versions of $W$. We wish to reduce this effect as much as possible. While compensating for quantization errors has been done before [14], we derive a tailored solution for LoQT.

During the merging update phase, we first dequantize to obtain $W_{T-1}$ and $P_{T-1}$, and then compute the update $W_T = W_{T-1} + P_{T-1}B_{T-1}$. This is immediately followed by re-quantizing to get $Q_T = q_{\text{NF4}}(W_T)$. Our goal is to minimize the quantization error $\|(Q_T + P_T B_T) - W_T\|$. Recall that $P_T$ is found based on the gradient and is not changed to compensate for the quantization error. Instead, we solve for $B_T$ in the merging step, initializing $B_T$ as $B_T \stackrel{\text{def}}{=} P_T^+(Q_T - W_T)$, where $P_T^+$ is the Moore-Penrose pseudo-inverse. This approach avoids initializing $B_T$ as zeros, as is commonly done [5], and instead uses it for minimizing the quantization error $\|Q_T - W_T\|$. We then iteratively refine $B_T$ over a maximum of five steps, by recomputing $Q_T = q_{\text{NF4}}(W_T - P_T B_T)$, improving the alignment between the full-precision $W$ and its quantized state.

As training advances and the learning rate decays, the magnitude of the update $B_{T-1}$ decreases. This leads to negligible differences $|q(Q_t + P_t B_t) - Q_t|$, which results in the loss plateauing early, as depicted in Fig. 4a. To address this, we implement an exponentially increasing scheduler for updating $W$. Drawing from the observation that the gradient rank decays exponentially (Lemma 3.1 in [8]), we start with an update interval $\tau$ and progressively increase the update intervals by a factor of $\psi$. The sequence of updates is then given by $(T_i)_{i=0}^{\infty} = (\tau + \psi^i)_{i=0}^{\infty}$. Each $T_i$ marks a training step $t$ when $W$ is updated. This scheduling ensures more frequent updates earlier in training and more well-spaced adjustments later, allowing for accumulation of sufficiently large gradients before each progressive update.

## 3 Experiments

We evaluate LoQT on language model pretraining by training LLaMA-based [15] language models on the C4 dataset [16], a collection of text in English that was extracted from the Common Crawl web-scrapes [16]. We train models of sizes of 60M, 130M, 350M, and 1B parameters, adhering to single-epoch training cycles determined by the Chinchilla Scaling Laws [17]. While LoQT is capable of training models up to 13 billion parameters on consumer GPUs, compute limits prevent us from training to convergence for sizes above 1B. We also benchmark LoQT on the GLUE test-suite for natural language understanding [18], the GSM8K [12] dataset for arithmetic reasoning and an Icelandic text dataset [10] to evaluate language adaptation via continued-pretraining. Runs were conducted on up to 4x 40GB NVIDIA A100s 2x 80GB NVIDIA H100s, or a single 24GB NVIDIA RTX 3090. The longest run was the training of the 1B models, taking approximately four days on the four A100s. The RTX 3090 was used for throughput and to empirically verify memory claims.

Table 1: Comparison of low-rank pre-training methods for LLaMA2-style language models on the C4 dataset. The table shows validation perplexity, memory estimates, and quantization states for LoQT. The rank ratio $r/d_{model}$ is relative to the largest weight matrix dimension. Perplexity values are averaged over three seeds showing mean and standard error. (*) Denotes results from GaLore [8]. Only one seed was used for the 1B experiment due to compute constraints.

|  | 60M | 130M | 350M | 1B |
|---|---|---|---|---|
| Full | $33.32 \pm 0.22$ (0.36G) | $24.51 \pm 0.03$ (0.76G) | $18.87 \pm 0.18$ (2.06G) | 15.56 (7.80G) |
| LoQT (Ours) | $33.98 \pm 0.15$ (0.23G) | $24.57 \pm 0.01$ (0.49G) | $19.12 \pm 0.01$ (0.98G) | 15.55 (3.16G) |
| LoQT-nq (No quant.) | $33.55 \pm 0.03$ (0.28G) | $24.37 \pm 0.02$ (0.63G) | $18.85 \pm 0.01$ (1.47G) | 15.20 (5.11G) |
| GaLore | $34.15 \pm 0.24$ (0.24G) | $24.81 \pm 0.04$ (0.52G) | $19.47 \pm 0.01$ (1.22G) | 15.64* (4.38G) |
| LoRA | 34.99* (0.36G) | 33.92* (0.80G) | 25.58* (1.76G) | 19.21* (6.17G) |
| ReLoRA | 37.04* (0.36G) | 29.37* (0.80G) | 29.08* (1.76G) | 18.33* (6.17G) |
| $r/d_{model}$ | 128 / 256 | 256 / 768 | 256 / 1024 | 512 / 2048 |
| Training Tokens | 1.1B | 2.2B | 6.4B | 13.1B |

We keep hyperparameters consistent across model sizes, with experiments conducted in BF16 format for memory efficiency. All models are trained with a maximum sequence length of 256, a total token batch size of 131K tokens, and a learning rate warmup for the first 10% of the training steps, followed by cosine annealing to 10% of the initial learning rate. Full experimental details, including the specific hyperparameters for each task, are provided in Appendix B.

**Baselines** For pretraining, we compare LoQT against LoRA [5], ReLoRA [7], GaLore [8], and a non-quantized version of LoQT, LoQT-nq. In our experiments, we apply these parameter-efficient training methods to the attention projection matrices and fully connected layers while maintaining full-rank embeddings. For the fine-tuning experiments, we compare LoQT against GaLore, LoftQ [14], LoRA, ApiQ [4], and LoQT-nq, or a subset thereof. All models that make use of update frequencies are trained using the same intervals, these are GaLore, ReLoRA, LoQT-nq, and LoQT. We start with an update interval of $T = 100$ and then exponentially increase the update frequency. This means that we do more frequent updates early and fewer as the model stabilizes (see § 4b for more details). A scaling parameter $\alpha = 0.5$ is used for LoQT and GaLore across all models, except for the 1B model where it is decreased to $0.25$. The same rank $r$ is used for all low-rank methods. All models are trained using the Adam optimizer, except GaLore which uses their GaLoreAdam optimizer for gradient projection. More details on hyperparameters are provided in the Appendix B.

### 3.1 Pretraining of Generative Language Models

Results and details of pretraining causal language models of sizes 60M, 130M, 350M, and 1B parameters are shown in Tab. 1. Model sizes are calculated based on the full models without any low-rank methods. We see that LoQT and LoQT-nq both perform very close to full rank pretraining and GaLore while using significantly less memory by keeping most of the model weights in a quantized state. For the 60M model, full training is only slightly better than LoQT, while we see results improve or stay within the standard error for the other sizes. We also notice a slight drop in performance from quantizing the original weight matrix, comparing LoQT and LoQT-nq. The key difference between the approaches is the theoretical memory estimates, e.g. where LoQT uses 59% less memory for the 1B model in full precision and 28% less memory than with GaLore.

Table 2: Results for LoQT, LoQT-nq, and GaLore using DeBERTaV3-base models on the GLUE development set. We report mean and standard error over three seeds. The best mean results on each dataset are shown in **bold**.

| Rank | Method | MNLI
Acc | QNLI
Acc | RTE
Acc | SST
Acc | MRPC
f1 | CoLA
Matt | QQP
f1 | STSB
PCorr | Average |
|---|---|---|---|---|---|---|---|---|---|---|
| 32 | LoQT-nq | 90.0±0.10 | 94.2±0.06 | **84.8±0.75** | **95.9±0.06** | 94.1±0.25 | **72.5±0.41** | **90.0±0.06** | 91.5±0.07 | **89.1** |
| 32 | LoQT | 90.0±0.09 | **94.3±0.04** | 84.1±0.91 | 95.5±0.10 | **94.4±0.20** | 70.5±0.35 | 89.2±0.02 | **91.5±0.13** | 88.7 |
| 32 | LoRA | 89.9±0.03 | 94.0±0.09 | 83.6±0.12 | 95.7±0.15 | 93.5±0.26 | 69.3±0.47 | 89.8±0.11 | 90.7±0.22 | 88.3 |
| 32 | LoftQ | 90.4±0.09 | 93.2±0.02 | 83.8±0.63 | 95.6 ±0.07 | 93.2±0.14 | 71.1±0.28 | 89.6±0.12 | 91.0±0.09 | 88.4 |
| 32 | GaLore | **90.3±0.07** | 94.0±0.04 | 83.7±0.79 | 95.6±0.07 | 93.4±0.38 | 70.7±0.24 | 89.8±0.05 | 90.6±0.01 | 88.5 |

## 3.2 Memory-Efficient Finetuning

We fine-tune the pretrained DeBERTa-V3-base[2] [19] model on the natural language understanding GLUE [11] tasks using LoQT and compare its performance with full fine-tuning baselines, LoRA, LoftQ, and GaLore. See Appendix 7 for details on hyperparameters. Results are given in Tab. 2.

We find that both LoQT-nq and LoQT perform well. And somewhat surprisingly, they sometimes surpass GaLore, LoftQ, and LoRA. This may indicate that initializing the LoRA factors with information about the gradient of $W$ is a beneficial starting point compared to standard initialization methods. Further experiments are needed to confirm and investigate these findings which we leave to future work.

**Arithmetic Reasoning on GSM8K**   We fine-tune quantized Llama-2 models (7B and 13B) on the GSM8K dataset [12] for arithmetic reasoning. As shown in Tab. 3, LoQT achieves average test set accuracies of $42.6\%$ and $52.9\%$ with the 7B and 13B models, respectively, performing comparably to other quantized fine-tuning approaches. Detailed hyper-parameters are provided in Appendix Tab. 8.

Table 3: GSM8K LLaMA-2 7B and 13B test accuracy with std. error. Best mean is in **bold**.

| Method | Bit | LLaMA-2-7B | LLaMA-2-13B |
|--------|-----|------------|-------------|
| LoRA | 16 | $41.7 \pm 0.3$ | $51.3 \pm 0.86$ |
| QLoRA | 4 | $41.9 \pm 0.2$ | $51.6 \pm 0.29$ |
| LoftQ | 4 | $41.9 \pm 0.9$ | $51.3 \pm 0.96$ |
| ApiQ | 4 | $42.1 \pm 0.5$ | $52.4 \pm 0.46$ |
| LoQT | 4 | $\mathbf{42.6} \pm 0.4$ | $\mathbf{52.9} \pm 0.12$ |

Table 4: Llama-7B fine-tuning on Icelandic. We report test set perplexity.

| Method | Perplexity $\downarrow$ |
|--------|------------|
| No training | 4.90 |
| Full | 3.79 |
| GaLore | 3.96 |
| LoQT-nq | **3.61** |
| LoQT | **3.63** |

**Continued Pretraining of Llama 7B**   We also evaluate LoQT on language adaptation of a large language model. We continue pretraining of the Llama-2-7B model using a curated subset of a public Icelandic text dataset extracted from [10] containing 770k documents. We compare LoQT with NF4 quantization, LoQT without quantization (LoQT-nq), regular training, and GaLore, using consistent hyper-parameters across all methods, results are shown in Tab. 4. LoQT achieves test set perplexity close to that of using full training or GaLore while reducing perplexity from 4.90 (non-trained model) to 3.63. Additional details are provided in Appendix C.1.

## 3.3 Memory and Throughput

**Memory Usage**   An overview of memory usage for GaLore, LoRA and LoQT is given in Tab. 5. We see that LoQT has the same number of parameters as LoRA for a given rank while using less memory for the optimizer states and gradients than in both LoRA and GaLore.

We compare LoQT to GaLore, the approach that gets closest in memory performance, for a model of size 13B in Fig. 1, and for other model-sizes in Fig. 6. We compare three different use cases, applying the methods on their own, combining them with an 8-bit Adam optimizer [20], and using per-layer weight updates with offloading (while still using 8-bit Adam). We see that LoQT significantly reduces the number of trainable parameters, and optimizer states, compared to GaLore.

Per-layer weight updates are essential for GaLore; without it, an additional ∼12 GB of VRAM is needed for the gradients of a 7B model, making full-parameter fine-tuning impossible on a 24GB GPU. Additionally, the per-layer gradient updates do not work with gradient accumulation. Using LoQT results in lower memory use than GaLore, even with per-layer gradient updates. When not using per-layer gradient updates, the difference becomes more pronounced as seen for the 7B model in Fig. 6.

LoQT enables training of 7B models without per-layer computations on a 24GB GPU, allowing for gradient accumulation and higher effective batch sizes. Our memory advantage allows for a batch size of 1280 tokens compared to GaLore's 256 for the 7B model on the 24GB RTX 3090. Using

---

[2]From `https://huggingface.co/microsoft/deberta-v3-base`.

Table 5: Comparison of memory usage for GaLore, LoRA, and LoQT. $W \in \mathbb{R}^{m \times n}$ ($m \leq n$), rank $r$.

|  | GaLore | LoRA | LoQT (Ours) |
|---|---|---|---|
| Weights | $mn$ | $mn + mr + nr$ | $mn + mr + nr$ |
| Optimizer States | $mr + 2nr$ | $2mr + 2nr$ | $2nr$ |
| Gradients | $mn$ | $mr + nr$ | $nr$ |
| Pretraining | Yes | No | Yes |
| Fine-Tuning | Yes | Yes | Yes |
| Quantizeable | No | Yes | Yes |

per-layer gradient updates, LoQT can train a 13B model on a single GPU. We refer to Fig. 8 in the Appendix for a comparison of how Adam, GaLore, and LoQT scale with increasing context length.

**Throughput**  We evaluate the throughput with a sample batch size of 16, and a total batch size of 512 using gradient accumulation, which is the largest power of two that fits on the GPU. We update the projection matrix $P$ for every 200 iterations. We evaluate the throughput using a 1B parameter model and rank 512 without per-layer gradient updates. We find that LoQT processes 16% fewer tokens per second than training with only AdamW, at 3996 tokens/s compared to 4782 tokens/s on the RTX 3090.

## 4  Ablations

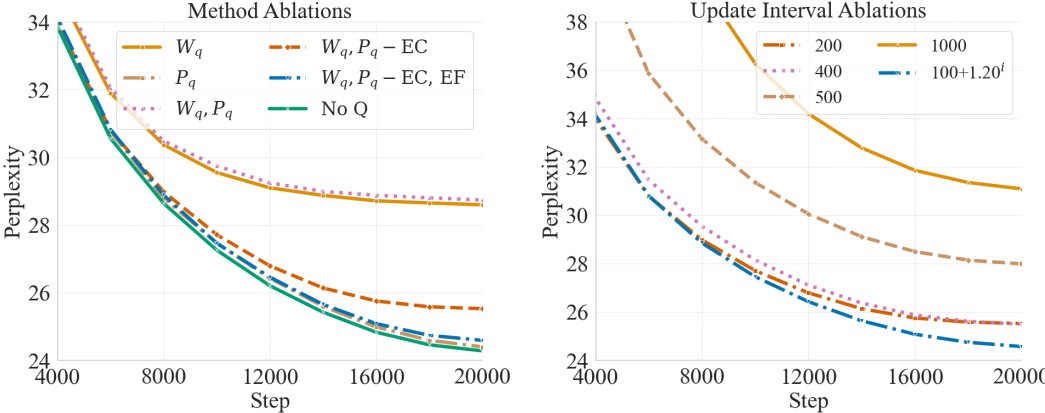

(a) EC: Error compensation, EF: Exp. increasing update interval.

(b) Ablation of update intervals: comparing fixed intervals to an exponentially increasing schedule.

Figure 4: Ablation results for update intervals, error-compensation, quantization using model size 130m, and rank 256. $W_q$: quantized $W$; $P_q$: quantized $P$; No Q: no quantization. The dynamic update interval $100 + 1.2^i$ grows exponentially for each step $i \in \mathbb{N}$.

**Quantization Error Compensation and Initialization**  We analyze the validation loss curves of 130 million parameter models to assess the impact of error quantization compensation. Fig. 4a shows that quantizing $W$, or both $W$ and $P$, without error compensation, or exponential interval updates leads to early stagnation of the loss. We also note that quantizing $P$ has a much smaller effect on the loss compared to quantizing $W$. Error compensation significantly improves the model's performance, resulting in approximately 3.5 points better perplexity. Adding exponentially increasing update intervals improves perplexity further by an additional 1.5 points, achieving performance close to that of models without quantization.

Without the quantization error compensation, detailed in §2.5, LoQT's performance stagnates earlier and diverges more from the other models. This demonstrates the effectiveness of our compensation approach in mitigating the quantization errors introduced during the update of $W$ with $PB$ and subsequent quantization steps.

**Projection Update Intervals** Our scheduling approach ensures more frequent updates earlier on in training when the weight adjustments are larger. As training progresses, the update intervals get larger, allowing for accumulating more updates to compensate for smaller changes at each step that might otherwise be canceled out by the quantization errors. Fig. 4b presents an ablation study of progressively increasing update intervals starting at 100 and increasing by a factor of $1.2^T$ up to 2500. We show the validation loss curves for fixed update frequencies 200, 400, 500, and 1000.

The results show that exponentially increasing the update interval is particularly beneficial for models employing quantization, enabling them to achieve the same perplexity as those without quantization. Conversely, the performance gains are more subtle for models that do not use quantization. We hypothesize that even these models might benefit from the larger projection interval intervals. This could be due to the reduction in the accumulation of errors from frequent updates of the projection factor $P$, as the influence of outdated optimizer statistics becomes less prevalent. Finally, an ablation on the ranks used for $P$ and $B$ is given in Fig. 5 in the Appendix.

## 5 Related Work

We now provide an overview of related work on quantization, parameter-efficient fine-tuning methods, and memory-efficient approaches.

### 5.1 Neural Network Quantization and NF4

Quantization compresses neural networks by converting high-precision values into lower-precision formats, significantly reducing storage requirements [21, 22, 23, 20]. The process involves taking a datatype of high precision, such as 32-bit, requiring 4 bytes of memory, and converting it into a representation with increasing rounding errors but lower memory cost. In this work, we use NF4 quantization [3], since it is a 4-bit code it only contains $2^4$ different values. NF4 works by first normalizing values onto the interval $[-1 : 1]$, these are then discretized onto quantiles of the normal distribution, $(q_i)_{i=1}^{16}$ (see [3] for details). The elements of a layer are divided into blocks of 64 weights. Each block $\beta$ has a scaling factor $\mathcal{M}_\beta = \max_{w \in \beta} |w_{32}|$.

$$w_{\text{NF4}} = q_{\text{NF4}}(w, \mathcal{M}_\beta) \tag{9}$$

$$\stackrel{\text{def}}{=} \text{argmin}_{q_i} |w/\mathcal{M}_\beta - q_i|, \tag{10}$$

$$w = q_{\text{NF4}}^{-1}(w_{\text{NF4}}, \mathcal{M}_\beta) \tag{11}$$

$$\stackrel{\text{def}}{=} \mathcal{M}_\beta \cdot w_{\text{NF4}}. \tag{12}$$

We provide an overview of different categories of quantization techniques, and how they relate to LoQT, in Appendix A. Compared to prior approaches, LoQT retains the benefits of reduced memory usage while minimizing accuracy loss, using high-precision updates on a low-rank representation. This allows for efficient model updates without the overhead of full matrix storage and re-quantization.

### 5.2 Adaptation of Pretrained Networks

Low-Rank Adaptation (LoRA) [5] enables fine-tuning of pretrained models using low-rank adaptors, effectively reducing the memory footprint by only training weight adaptors for targeted layers. However, simple low-rank training using LoRA factor matrices has not been shown to work for pretraining [7].

LoRA employs trainable low-rank matrices $A$ and $B$ that are used to update $W$ following $W_t = W_{t-1} + AB$, where $W_{t-1}$ is frozen to enable precise adjustments within a low-rank framework. Since LoRA only trains $A$ and $B$ and keeps $W$ fixed, QLoRA [5] explore quantizing $W$. They fine-tune a quantized model $q(W) = W_q$ with 4-bit precision using randomly initialized 16-bit precision factors $A$ and $B$. To address quantization errors $\mathcal{E} = |W_q - W|$, low-rank factors of the quantization error $\mathcal{E}$ have been used [14].

LoQT extends LoRA to both pretraining and fine-tuning. Unlike traditional LoRA, LoQT uses $A$ and $B$ to refine $W$ throughout training, with $A$ initialized from $W$'s gradient projection and $B$ trained along this gradient path. LoQT also incorporates quantization and targeted optimization iterations

similar in spirit to LoftQ [14], correcting for quantization errors in $W_q$, thus better aligning it with the original non-quantized $W$.

### 5.3 Memory Efficient Optimization

**Optimizer memory consumption**   A significant portion of the memory needed to train neural networks is typically consumed by optimizer states. Notably, Adam [9], one of the most widely used optimizers, uses double the amount of memory as the gradient matrix to maintain first and second-order gradient statistics. Efforts to reduce this overhead have led to the development of adaptive optimization algorithms like Adafactor [24], which achieves sub-linear memory costs by factorizing the second-order statistics into a row-column outer product. GaLore [8] expands on this concept by using low-rank factorization and projecting low-rank gradients up to full size when updating model weights.

**Periodic updating of weight matrices**   ReLoRA [7] combines low-rank updates with initial full-rank training. They find that doing one-third of the training in full-rank, and the subsequent two-thirds in low-rank (see §5.2) results in comparable performance to standard training methods.

**Low-rank Gradients**   GaLore [8], focuses on the structure of the gradients, projecting them into a low-rank space using factors $P$ and $Q$, which are derived from a truncated singular value decomposition (SVD) of the weight matrix gradient, $G_W \approx P_r \Sigma_r Q_r$. This reduces memory costs associated with storing the optimizer states and aligns with findings from recent studies which suggest that learning primarily occurs within a low-dimensional subspace at a given time [25, 26]. This can be further combined with applying per-layer gradient updates, reducing the memory needed for storing the gradients for the full model at once [13].

LoQT builds on GaLore's gradient projection (§2.1) to initialize LoRA factors while updating the full matrix following a schedule inspired by ReLora, while only training one low-rank matrix per layer. We achieve comparable quality to GaLore and better performance than ReLoRA while reducing tunable parameters and memory usage compared to both approaches.

## 6   Discussion and Conclusion

We have presented LoQT, a method for memory-efficient pretraining and adaptation of quantized models. Key insights behind the approach are the benefits of initializing low-rank factors using the gradient of the weight matrix and using exponentially increasing update gaps that make updating of a quantized model feasible. While our initial goal was to lower memory usage, to facilitate the training of models such as LLMs on consumer-grade hardware, we are also cautiously excited about the results sometimes exceeding those of the baselines. We hope to see this explored in more detail in future work.

Our method is general and has the potential to open up new ways of decreasing memory use and improving training throughput through further optimization of our implementation. This could be done by using other quantization methods such as NF2 [3] or quantization of activations, making it possible to do the matrix multiplications using modern tensor core formats such as FP8 or INT4.

## 7   Impact and Limitations

The presented work has the potential to benefit those working in hardware-constrained settings by enabling more efficient training on consumer-grade hardware. We are particularly excited to see the method being applied in single GPU settings.

We have validated LoQT on several model sizes, by training over many steps, by fine-tuning on a standard benchmark for natural language understanding, mathematical reasoning, and language adaptation. While we are confident in our results, further exploration of training duration, data diversity, and hyper-parameter tuning might lead to different results in those settings and we encourage users to confirm the benefit of LoQT for their approach.

## 8 Acknowledgements

This work is supported by the Danish Data Science Academy, which is funded by the Novo Nordisk Foundation (NNF21SA0069429) and VILLUM FONDEN (40516). Serge Belongie and Vésteinn Snæbjarnarson are supported by the Pioneer Centre for AI, DNRF grant number P1. MJK acknowledges support from the Carlsberg Foundation and the Novo Nordisk Foundation. Mads Toftrup gratefully acknowledges the Data-Intensive Systems research group at Aarhus University for providing GPU access.

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

# A Quantization Methods

Quantization methods can be broadly categorized into Quantization-Aware Training (QAT), Post-Training Quantization (PTQ), and Fully Quantized Training (FQT).

**Quantization-Aware Training (QAT)**  QAT [27, 28, 29, 30, 2] integrates quantization in the training process by emulating inference time quantization where the model weights are quantized. By maintaining high precision gradients and optimizer states, QAT allows the model to adapt to quantized weights while preserving accuracy. These methods predominantly focus on weight-only quantization approaches, which involve converting weight matrices into low-precision formats and then upcasting them just before computation [30, 2]. This allows the main computation to occur at high precision, effectively preserving model accuracy while significantly compressing the model [31]. However, QAT can require significant computational resources due to the need for full precision gradient calculations and large optimization states [3].

**Post-Training Quantization (PTQ)**  PQT [31, 32, 33, 34, 35, 20, 36, 37, 38] involves converting a pretrained high-precision model into a lower precision format. This can be done directly or by using a subset of the training data to calibrate the quantization process or fine-tune the quantized weights to adapt the model to the quantization. However, PTQ often results in reduced accuracy compared to QAT because the model does not learn to adjust to quantized weights during training [31, 34].

**Fully Quantized Training (FQT)**  FQT aims to minimize memory and accelerate the training process by quantizing both forward and backward passes [39, 40, 41, 42, 43]. These methods often require specialized hardware [44, 45] but are promising for efficient training, and current approaches cannot maintain accuracy [45].

LoQT is a form of QAT that gets close to FQT. As we perform a variant of LoRA (see §5.2), we factor the layers $W$ into two matrices $P$ and $B$. We quantize the $W$ and $P$ with NF4, but keep $B$ in 16-bit precision. We periodically update the $W$ matrices using the product of the fixed $P$ and the updated $B$s without ever dequantizing it all at once, only layerwise when merging in $PB$. This approach retains the benefits of reduced memory usage while minimizing accuracy loss, focusing high-precision updates on a low-rank representation, and allowing efficient model updates without the overhead of full matrix re-quantization.

**Choice of Quantization Method**  We chose NF4-quantization because it has been shown to work well [46, 3]. Unlike other methods that adapt low-rank factors to quantization, such as LoftQ [14] and LQLoRA [47], LoQT operates under different constraints. Specifically, we do not have the flexibility to freely choose both $A$ and $B$ factors because the matrix $A$ is already fixed as the projection matrix $P$ containing gradient information. Both LoftQ [14] and LQLoRA [47] use the SVD of the quantization error to initialize the low-rank factors $A$ and $B$, aiming to minimize the difference between the quantized $W$ and $Q + AB$. The SVD gives the factorization $U\Sigma V^T$, where the top $r$ singular vectors in $U$ and $V$ are used to initialize the low-rank factors $A$ and $B$, respectively.

In contrast, LoQT takes a different approach due to the fixed nature of our low-rank adapter $P$ (analogous to $A$ in LoftQ and LQLoRA). Instead of applying SVD to the quantization error, we aim to minimize an objective where $W$, $Q$, and $P$ are fixed. We derive $B$ using the formula $B = P^+(W - Q)$, where $P^+$ is the Moore-Penrose pseudo-inverse of $P$ rather than the inverse. Incorporating information about the diagonal approximation of the Fisher information matrix into our objective could potentially reduce the error even further, a direction we are interested in exploring in future work.

# B Hyperparamters

We provide the hyperparameter configurations and setups used in our experiments to facilitate the reproduction of all results presented in this paper.

## B.1 Pretraining

For the pretraining results shown in Tab. 1, we adopted configurations from GaLore [8] and tested pretraining methods on different LLaMA 2 model sizes using the C4 dataset. Training was conducted with optimizer states in BF16 precision, and NF4 precision quantization was used for LoQT. The model rank was adapted based on the largest layer with specific parameters.

Tab. 1 shows the ratio $r/d_{model}$, which denotes the rank relative to the largest weight matrix dimension. All experiments used a maximum sequence length of 256, learning rate warmup for the first 10% of training steps, and cosine annealing for the learning rate schedule, decaying to 10% of the initial rate. Galore, LoQT-nq, and LoQT used exponentially increasing update frequencies starting at 100 and increasing by $100 + \psi^i$, where $\psi$ is 1.2 and $i$ is the update counter (see Section C.1 for more details).

We tested learning rates of 0.01, 0.005, 0.001, and 0.0005 across different model sizes. For models ranging from 60M to 350M parameters, a learning rate of 0.01 yielded the best performance. In contrast, full-rank models required smaller learning rates: 0.001 for 60M to 350M models and 0.0005 for the 1B model. To scale the learning rates for LoQT, LoQT-nq, and Galore, we employed a scale parameter $s$ set to 0.5 and 0.25 for the 1B model. This parameter functions similarly to the LoRA alpha parameter, determining the weight of the learned factors for LoQT and LoQT-nq. For Galore, our experiments indicated that $s = 0.5$ was more effective than the 0.25 reported in [8]. This scaling approach effectively adjusts the learning rate, resulting in an actual rate of 0.005 for the multi-head attention and feed-forward layers in LLaMA models, which is relatively large compared to the 0.001 used for full-rank models. Higher learning rates led to spikes in the training loss for both full-rank and LoQT models.

Table 6: Pretraining hyperparameters of LLaMA models for evaluation. (-) Indicates we did not train such a model.

| Model Size | Hidden/Intermediate | Attention Heads | Layers | Steps | Data Amount | Rank |
|---|---|---|---|---|---|---|
| 60M | 512 / 1376 | 8 | 8 | 10K | 1.3B | 128 |
| 130M | 768 / 2048 | 12 | 12 | 20K | 2.6B | 256 |
| 350M | 1024 / 2736 | 16 | 24 | 60K | 7.8B | 256 |
| 1B | 2048 / 5461 | 24 | 32 | 100K | 13.1B | 512 |
| 7B | 4096/11008 | 32 | 32 | - | - | 1024 |
| 13B | 5120/13824 | 40 | 40 | - | - | 1536 |

## B.2 Fine-tuning

We test learning rates in the range of $1 \times 10^{-5}$ to $5 \times 10^{-4}$. For LoQT and LoftQ, we employed normal float NF4 quantization and performed five iterations of optimizing the error of quantization. We used a batch size of 32 and a maximum sequence length of 256. Tab. 7 summarizes the detailed hyperparameters for tasks in GLUE using the DeBERTaV3-base model. We use a fixed projection gap of 2400 for all runs. Each of the parameter-efficient training methods is applied to all linear layers of the network, including attention projection and feed-forward layers, while the embedding layer is not trained.

Table 7: Hyperparameter setup for LoQT-nq, LoQT, LoftQ[14], LoRA[14], and Galore across various tasks on the GLUE benchmark.

| Method | Hyper-parameter | MNLI | RTE | QNLI | MRPC | QQP | SST-2 | CoLA | STS-B |
|---|---|---|---|---|---|---|---|---|---|
| LoQT, LoFTQ | # Epochs | 5 | 20 | 10 | 60 | 10 | 10 | 20 | 60 |
| | Learning Rate | $1 \times 10^{-4}$ | $5 \times 10^{-5}$ | $5 \times 10^{-5}$ | $1 \times 10^{-4}$ | $5 \times 10^{-5}$ | $5 \times 10^{-5}$ | $1 \times 10^{-4}$ | $5 \times 10^{-5}$ |
| LoRA, Galore | # Epochs | 10 | 30 | 30 | 30 | 30 | 30 | 30 | 30 |
| | Learning Rate | $1 \times 10^{-5}$ | $2 \times 10^{-5}$ | $1 \times 10^{-5}$ | $2 \times 10^{-5}$ | $1 \times 10^{-5}$ | $2 \times 10^{-5}$ | $2 \times 10^{-5}$ | $3 \times 10^{-5}$ |

## C Rank Ablation

Fig. 5 shows the validation perplexity versus training steps for various ranks using LoQT-nq and LoQT on a 130 million parameter model over 20,000 iterations. All models employ an exponentially increasing update frequency starting at 100, with a factor of $1.2^{T_i}$. The results demonstrate that both the quantized (LoQT) and non-quantized (LoQT-nq) models follow a similar trajectory for ranks ranging from 64 to 512. However, for the smaller rank of 64, there is a slight divergence between LoQT-nq and LoQT, indicating a limit to how low the rank can be while maintaining accuracy with quantization. This plot highlights the tradeoff between rank and perplexity, suggesting that while our method supports low-rank training, there is a minimum rank threshold needed to achieve results comparable to regular pretraining.

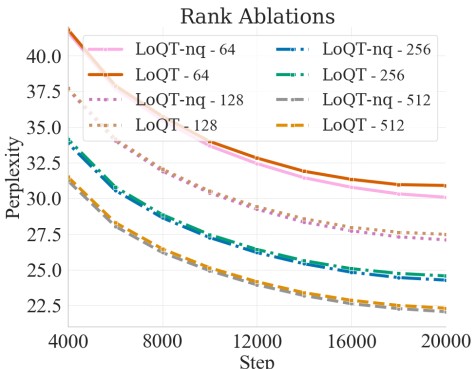

Figure 5: Rank ablation for LoQT and LoQT-nq showing perplexity as a function of steps.

### C.1 Memory Measurements

Fig. 6 demonstrates that LoQT requires less memory than GaLore and Adam, even without using per-layer gradients [13] or Adam 8-bit [20]. The gap between LoQT and the baselines increases with larger model sizes. The configurations and ranks for each model are shown in Tab. 6. With LoQT and Adam 8-bit, it is possible to pretrain a 13B model with rank 1024 on a GPU with 24GB of VRAM. This enables training with LoQT on consumer GPUs, such as the NVIDIA 4090, using a small per-GPU token batch size of 256. Fig. 1 in the main text provides a detailed breakdown of each memory component for the 13B model. Maximum memory allocated is measured using `nvitop` (https://github.com/XuehaiPan/nvitop).

**Finetuning without merging low-rank factors**    Tab. 9 shows how LoQT and LoQT-nq perform when not using the merging of factors while training. We see that the results are slightly worse than those where merging is performed; however, the results are still better than LoftQ, LoRA, and Galore, showing that in the finetuning case, where we already have a pretrained model, we can omit to merge low-rank factors into W, and still get good results. This would allow W to be kept fixed and quantized while training only the adapters, which enables using LoQT like LoRA, where multiple adapters can be trained for the same set of frozen weights.

**Task adaptation for GSM8K on Llama 7B and 13B**    The GSM8K dataset [12] "is a dataset of 8.5K high-quality linguistically diverse grade school math word problems created by human problem writers. The dataset is segmented into 7.5K training problems and 1K test problems". It has been used extensively to benchmark LLMs and is a good candidate for evaluating LoQT. We experimented with 7B and 13B models, performing a hyperparameter search to find the optimal learning rate for each method. Using the best learning rate, we trained each model over three seeds for three epochs with a sequence length of 512, applying 4-bit quantization for fine-tuning Llama-2 models on the GSM8K training set. We report the average test set accuracy and standard error in Tab. 3. LoQT achieves an accuracy of 42.6 for Llama-7B and 52.9 for Llama-2 13B. Both results are average obtained over three seeds without merging and with rank 64. Tab. 8 lists the hyper-parameters. We evaluate the fine-tuned and quantized model on the validation set and report the best perplexity across learning rates in Tab. 8. For each of the methods, only the attention projection matrices are trained.

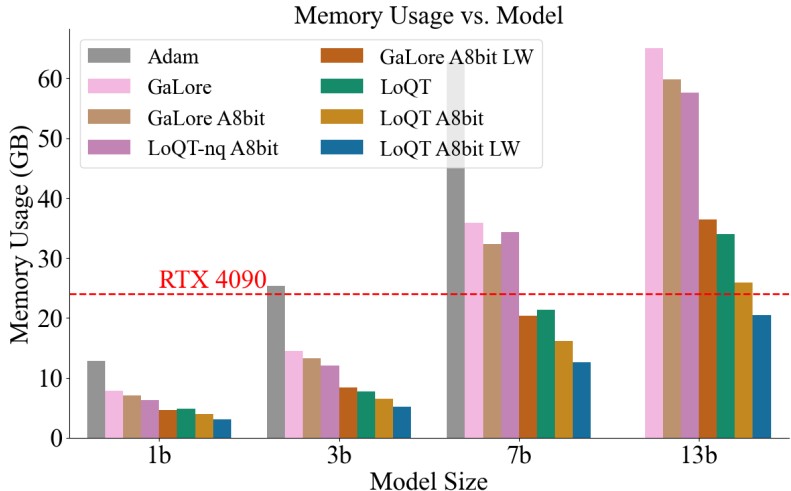

Figure 6: Memory usage for LoQT vs baselines for different model sizes. LW means per-layer gradient updates as per [13], and A8bit means with Adam 8-bit. We evaluate using a token batch size of 256.

Table 8: Hyper-parameters used for the GSM8K task.

| Hyper-parameter | Value |
|---|---|
| Optimizer | AdamW |
| Weight decay | 0.1 |
| LR | $\{0.1, 0.5, 0.7, 1, 3, 4\} \times 10^{-4}$ |
| LR scheduler | cosine |
| Warmup ratio | 3% |
| Epochs | 3 |
| Batch size | 16 (7B), 8 (13B) |
| Max sequence length | 512 |

**Continued pretraining of Llama 7B**  We run our pretraining experiments for models up to 1B parameters. To demonstrate the feasibility of LoQT on a larger model size in a practical application, we do continued pretraining for language adaptation. To this end, we start with the `meta-llama Llama-2-7b-hf` from Hugging face and run 1500 updates (1 epoch) on the model using a batch size of 512. We compare LoQT with NF4 quantization, LoQT-nq (no quantization), regular training, and GaLore. We use rank 1024 for all models where applicable, adam8bit optimizer, and 150 warmup steps. For LoQT and GaLore we use a learning rate of 0.0002 with a scale factor of 0.25 and for the others a learning rate of 0.00005. The dataset we train on is a curated subset of a public Icelandic text dataset [10] with ~770k documents and a corresponding evaluation set. We released the data splits at (`https://huggingface.co/datasets/vesteinn/loqt_icelandic`). We chose Icelandic since the model has a limited performance on the language yet it was included to some degree in the pretraining data, enabling a clear improvement trajectory. The results comparing Galore, Regular training (Full), and LoQT are shown in Tab. 4. LoQT and LoQT-nq perform the best at 3.61 and 3.63 in perplexity, similar to full training at 3.79, while GaLore gets 3.96 and the original model 4.90.

## D   Memory-Saving Ablations

To evaluate the differences between memory savings with layer-wise gradient computations and 8-bit Adam, we conduct an ablation experiment using a 130M parameter model. We compare four settings: regular training, 8-bit Adam, layer-wise gradient updates, and a combination of 8-bit Adam with layer-wise updates, tested on Galore, LoQT, and regular FP16 training. Our results, illustrated in

Table 9: Results with LoQT, LoQT-nq, and GaLore of DeBERTaV3-base models on the GLUE development set. We report mean and standard error over three seeds. The best results on each dataset are shown in **bold**. "No-Merge" means we do not update the pretrained matrix $W$ during training.

| Rank | Method | MNLI Acc | QNLI Acc | RTE Acc | SST Acc | MRPC f1 | CoLA Matt | QQP f1 | STSB PCorr | Average |
|------|--------|------|------|------|------|------|------|------|------|---------|
| 32 | LoQT-nq | 90.0±0.10 | 94.2±0.06 | 84.8±0.75 | **95.9±0.06** | 94.1±0.25 | **72.5±0.41** | **90.0±0.06** | 91.5±0.07 | **89.1** |
| 32 | LoQT | 90.0±0.09 | **94.3±0.04** | 84.1±0.91 | 95.5±0.10 | **94.4±0.20** | 70.5±0.35 | 89.2±0.02 | 91.5±0.13 | 88.7 |
| 32 | LoQT-nq - no Merge | 90.0±0.10 | 94.1±0.01 | 84.5±0.01 | 95.6±0.03 | 93.8±0.01 | 72.0±0.01 | 89.8±0.01 | **91.6±0.01** | 89.0 |
| 32 | LoQT- no Merge | 90.0±0.12 | 94.1±0.01 | **86.1±0.15** | 95.7±0.02 | 94.2±0.01 | 71.4±0.20 | 89.6±0.01 | 90.8±0.01 | 88.9 |
| 32 | LoRA | 89.9±0.03 | 94.0±0.09 | 83.6±0.12 | 95.7±0.15 | 93.5±0.26 | 69.3±0.47 | 89.8±0.11 | 90.7±0.22 | 88.3 |
| 32 | LoftQ | 90.4±0.09 | 93.2±0.02 | 83.8±0.63 | 95.6±0.07 | 93.2±0.14 | 71.1±0.28 | 89.6±0.12 | 91.0±0.09 | 88.4 |
| 32 | GaLore | **90.3±0.07** | 94.0±0.04 | 83.7±0.79 | 95.6±0.07 | 93.4±0.38 | 70.7±0.24 | 89.8±0.05 | 90.6±0.01 | 88.5 |

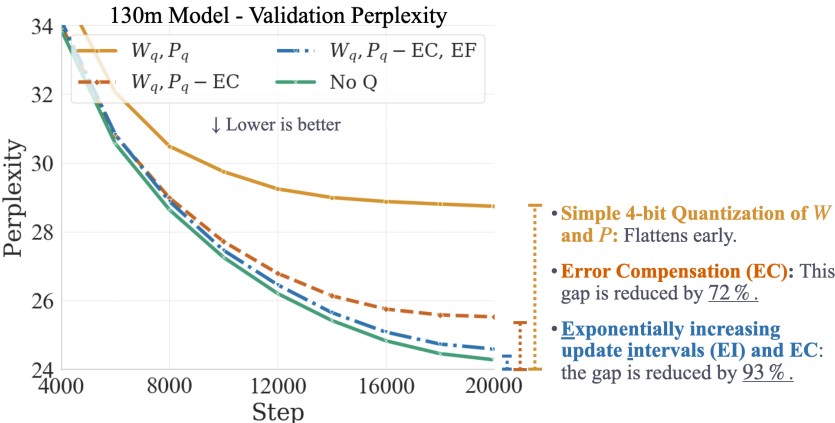

Figure 7: Naive Quantization of W and P vs including Error Compensation(EC) and Exp. increasing intervals(EI).

Fig. 9 show that adding these memory-saving components introduces a small decrease in performance. Importantly, LoQT experiences a proportionally smaller decrease in performance compared to Galore and full training when combining both 8-bit and layer-wise updates. These results demonstrate that while memory savings come with some trade-offs, LoQT maintains good performance. In addition, due to the lower memory requirements of LoQT, we enable training of larger models without using layer-wise gradient computations and 8-bit Adam.

**Memory savings with varying sequence lengths** With larger contexts, the overall memory consumption is increasingly influenced by activations. Following prior work [8, 48], our experimentation has focused on the setting of shorter context lengths (256 tokens). But as demonstrated in Fig. 8, the benefit of LoQT does transfer to longer context lengths, enabling training of Llama 7B on consumer hardware with a context length of 2048 and 4096 on a 40GB A100 without activation checkpointing.

# E   Generalization to other architectures and models

LoQT should work with any type of deep neural network using linear layers, such as vision transformers or state space models. To narrow the scope of our work and provide a more detailed analysis, however, we choose to focus on a well-studied auto-regressive language model and a bi-directional masked language model that has been commonly used as a basis in much of the related work. We hope to see LoQT being used for other model architectures.

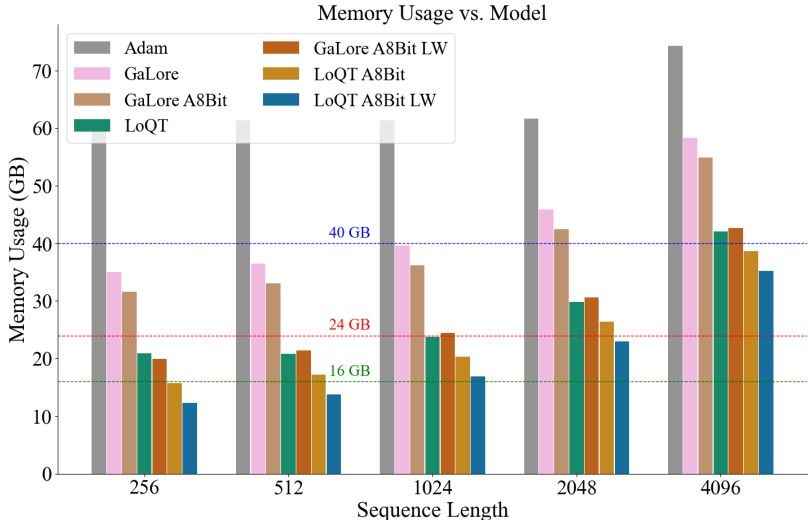

Figure 8: Memory usage for common context lengths (256 to 4096) for the LLaMA 7B model measured using `torch.cuda.max_memory_allocated`. We include lines representing 16GB, 24GB, and 40GB VRAM limits to indicate which configurations fit within the VRAM capacities of standard NVIDIA GPUs.

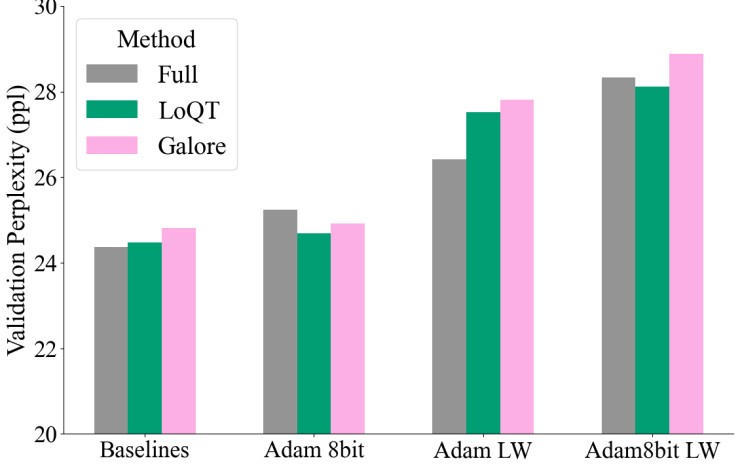

Figure 9: Validation perplexity with various optimization techniques: Adam 8bit, per-layer updates, and their combinations, compared to baseline training without these optimizations. LW means per-layer gradient updates as per [13], and Adam8bit means with Adam 8-bit.

