# OpenReview forum: "LoQT: Low-Rank Adapters for Quantized Pretraining"
_NeurIPS.cc/2024/Conference — NeurIPS 2024 poster_

### Official Review · Reviewer_Ekw9 · 2024-07-08

**Soundness:** 3
**Presentation:** 3
**Contribution:** 2
**Rating:** 4
**Confidence:** 3

**Summary:**

This paper proposes LoQT, a new method for efficiently training quantized models. LoQT is based on the low-rank decomposition of gradient matrices, and is inspired from GaLore [1] and LoRA [2] methods for transformers architectures. The work’s fundamental contributions are: 1) Propose a low-rank factors initialization (for each pre-trained matrix) based on gradient matrix and (weight) quantization error. 2) Show that Low-rank Gradients can be seen as (low-rank) adapters.
The numerical experimental results of the proposed method outperform other baselines on the GLUE task.


---

[1] Jiawei Zhao, Zhenyu Zhang, Beidi Chen, Zhangyang Wang, Anima Anandkumar, and Yuandong Tian. Galore: Memory-efficient llm training by gradient low-rank projection, 2024.

[2] Edward J. Hu, Yelong Shen, Phillip Wallis, Zeyuan Allen-Zhu, Yuanzhi Li, Shean Wang, Lu Wang, and Weizhu Chen. Lora: Low-rank adaptation of large language models, 2021.

**Strengths:**

1) LoQT successfully addresses the optimizer states memory concern for large neural networks training, by employing a low-rank decomposition of the gradient matrices. Different from GaLore [1], LoQT also quantizes (with NF4 from QLoRA [2]) the weights. LoQT successfully takes into account the induced quantization errors, when updating the low-rank factors.
2) The authors show that GaLore strategy can be formulated into a low-rank adapters decomposition, and that it is sufficient to train (with gradient descent) one of the low-rank factors.
3) This paper proposes a new periodically update strategy for fine-tuning pre-trained weights $W_{t+1} = W_t + P_t B_t$, with no need of optimizer states of the (full-rank) weights: only the optimizer states of $B_t$ are required.

---

[1] Jiawei Zhao, Zhenyu Zhang, Beidi Chen, Zhangyang Wang, Anima Anandkumar, and Yuandong Tian. Galore: Memory-efficient llm training by gradient low-rank projection, 2024.

[2] Tim Dettmers, Artidoro Pagnoni, Ari Holtzman, and Luke Zettlemoyer. Qlora: Efficient finetuning of quantized llms, 2023.

**Weaknesses:**

1) LoQT is mainly based on GaLore [1] with an additional step of weight quantization. Different from GaLore, no mathematical analysis is proposed, and the projection steps (SVD) are identical to GaLore's.
2) The main novelty of LoQT is its initialization that compensates the quantization error induced from the weight quantization. But (i) no contribution about weight quantization is proposed (NF4 strategy from QLoRA [2] is reused), and (ii) no discussion with other strategies that adapt low-rank factors to quantization (ApiQ [3], Loftq [4], LQLoRA [5]) is proposed.
3) The experiments again strictly follow the benchmark of GaLore [1], no experiments on 'standard' LLMs (such as LLaMA-7B; see for e.g. [5]) is given to fairly assess the computational gain of LoQT.


---

[1] Jiawei Zhao, Zhenyu Zhang, Beidi Chen, Zhangyang Wang, Anima Anandkumar, and Yuandong Tian. Galore: Memory-efficient llm training by gradient low-rank projection, 2024.

[2] Tim Dettmers, Artidoro Pagnoni, Ari Holtzman, and Luke Zettlemoyer. Qlora: Efficient finetuning of quantized llms, 2023.

[3] Liao, B. and Monz, C. (2024). Apiq: Finetuning of 2-bit quantized large language model. arXiv preprint arXiv:2402.05147.

[4] Li, Y., Yu, Y., Liang, C., Karampatziakis, N., He, P., Chen, W., and Zhao, T. (2023). Loftq: Lora-fine-tuning-aware quantization for large language models. In The Twelfth International Conference on Learning Representations.

[5] Guo, H., Greengard, P., Xing, E., and Kim, Y. (2023). Lq-lora: Low-rank plus quantized matrix decomposition for efficient language model finetuning. In The Twelfth International Conference on Learning Representations.

**Questions:**

1) Can you explain the difference between LoQT error compensation mechanism, and already published strategies such as LQLoRA [1] ?
2) In appendix (Table.4 and Figure.6) you have run LoQT on a LLaMA with 7B parameters, can you provide the corresponding perplexity results ? Is it competitive with the corresponding literature (ApiQ [2], Loftq [3], LQLoRA [1])?
3) Section 3.3 proposes to update weights $W_t$ (by merging low-rank factors). How does LoQT perform, when no update is realized on the pre-trained weights $W$? In particular, for fine-tuning on 10 different tasks LoQT would require 10 different models, while LoRA strategies would require a single full-rank model and 10 (smaller) adapters.
4) (minor) l.139 ``From (5) '': I think there is a typo regarding the ref.

---

[1] Guo, H., Greengard, P., Xing, E., and Kim, Y. (2023). Lq-lora: Low-rank plus quantized matrix decomposition for efficient language model finetuning. In The Twelfth International Conference on Learning Representations.

[2] Liao, B. and Monz, C. (2024). Apiq: Finetuning of 2-bit quantized large language model. arXiv preprint arXiv:2402.05147.

[3] Li, Y., Yu, Y., Liang, C., Karampatziakis, N., He, P., Chen, W., and Zhao, T. (2023). Loftq: Lora-fine-tuning-aware quantization for large language models. In The Twelfth International Conference on Learning Representations.

**Limitations:**

Impact and limitations properly discussed.

---

> ### Author Rebuttal · Authors · 2024-08-07
>
> We thank the reviewer for their thorough feedback. We appreciate the recognition of LoQT’s efficient handling of optimizer state memory, its effective quantization error management and its periodic update strategy.
>
> ## 1. Additional Finetuning Experiments
> Excellent suggestion; we agree that more fine-tuning experiments should be included. We refer to the common comment where we describe fine-tuning of Llama 7B for a downstream task and continue pretraining for adaptation of the model to a new language. In both cases, LoQT excels.
>
> ## 2. LoQT is mainly based on GaLore with weight quantization
> While our approach combines elements from existing methods like GaLore and others (see our detailed response to CPTS), we believe LoQT introduces significant new contributions, drawing on many related works besides GaLore in a non-trivial manner. We are the first to show that LoRA works well for pretraining given the right initialization, opening up further exploration in the LoRA literature. To our knowledge, we are also the first to present a method that can perform memory-efficient pretraining with most weights in 4-bit precision, resulting in a model that is already in 4-bit precision and thus does not require post-training quantization or quantization-aware training afterwards.
>
> We show, theoretically, that GaLore can be rewritten to a LoRA format, where one of the low-rank factors is initialized as the gradient projection matrix P. We then prove that only updating B and keeping P and W frozen results in the same gradient updates as Galore. This enables a separation that enables quantization. However, we see that naive quantization of W and P leads to a significant decrease in quality (See Figure 1 in extra PDF). To tackle this, we developed a tailored error compensation approach, which is described in our response below. We also saw that during the later part of training, the errors we get from the quantization increasingly affect the loss. We show that less frequent updates over time help because we do fewer quantizations of W and P. This also speeds up training since it requires fewer SVDs during training.
>
> Without the clear separations of frozen and trainable weights that enable us to use quantization, and without error compensation initialization of $B$ and without exponentially increasing update intervals, our method would not perform comparably to full training.
>
> ## 3. Quantization and Related Work
> We chose NF4 simply because it has been shown to work well. Our goal is not to introduce a new way of quantization nor to thoroughly compare to the post-training quantization literature. However, we appreciate the suggestion of considering LoQT as a post-training quantization method and will consider this in more detail in the final manuscript.
>
> Unlike other methods that adapt low-rank factors to quantization, such as LoftQ and LQLoRA, LoQT operates under different constraints. Specifically, we do not have the flexibility to freely choose both $A$ and $B$ factors because the matrix $A$ is already fixed as the projection matrix $P$ containing gradient information.
>
> We will include a more thorough discussion and comparison of ApiQ, LoftQ, and LQLoRA in the final manuscript. See also our answer below for a direct comparison.
>
> ## 4. Difference between the LoQT error compensation mechanism and already published strategies such as LQLoRA and LoftQ
>
> Both LoftQ and LQLoRA use the SVD of the quantization error to initialize the low-rank factors $A$ and $B$, aiming to minimize the difference between the quantized $W$ and $Q + AB$. The SVD gives the factorization $U \Sigma V^T$, where the top $r$ singular vectors in $U$ and $V$ are used to initialize the low-rank factors $A$ and $B$, respectively.
>
> In contrast, LoQT takes a different approach due to the fixed nature of our low-rank adapter $P$ (analogous to $A$ in LQLoRA). Instead of applying SVD to the quantization error, we aim to minimize an objective where $W$, $Q$, and $P$ are fixed. We derive $B$ using the formula $B = P^+(W - Q)$, where $P^+$ is the Moore-Penrose pseudo-inverse of $P$ rather than the inverse. Incorporating information about the diagonal approximation of the Fisher information matrix (like in LQLoRA) into our objective could potentially reduce the error even further, a direction we are interested in exploring in future work. Additionally, LQLoRA proposes dynamically adjusting the bit-width.
>
> ## 5. Clarification on 7B LLaMA Experiments
>
> We did not pre-train a 7B LLama model, as mentioned in the table caption of Table 4.
> We understand this is confusing, but we opted to include them for reference regarding the memory experiments presented in Figures 1 and 6.
> However, we have provided new results for continued pre-training on a 7B model, as described in the **common response**.
>
> ## 6. Finetuning without merging low-rank factors
> Good suggestion. We have done this experiment showing how LoQT and LoQT-nq perform when not using the merging of factors while training. The results are shown in Table 1 in the extra PDF indicated with (No-Merge). We see that the results are slightly worse than those where merging is performed; however, the results are still better than LoftQ, LoRA, and Galore, showing that in the finetuning case, where we already have a pretrained model, we can omit to merge low-rank factors into W, and still get good results. This would allow W to be kept fixed and quantized while training only the adapters as suggested, at least for some tasks.

---

> > ### Comment · Reviewer_Ekw9 · 2024-08-12
> >
> > Thank you for the rebuttal. Regarding the capabilites of LoQT wrt fine-tuning, it would be fair to compare to ''current'' standard fine-tuning benchmarks, i.e. C4 and/or Wikitext datasets with a LLaMA quantized with 3 bits (or lower; see LQLoRA for e.g.), but I can understand time is limited for the rebuttal. Overall, the authors provided detailed answers, I will increase my score.

---

> > > ### Author Response · Authors · 2024-08-12
> > >
> > > We thank the reviewer for raising their score. We would, however, like to emphasize strongly that we are not presenting our work as a post-training quantization technique, but rather as a method to save resources during pre-training and fine-tuning while still providing competitive performance. We have provided evaluations using well-acknowledged benchmarks for pre-training and fine-tuning to support this focus as recognized by the other reviewers.
> > >
> > > While we recognize the work could also be considered more strongly from the post-training quantization literature, it is simply not what we have done and an orthogonal direction to quantized pre-training and significant memory reduction. We fully acknowledge this distinction in the final version of the manuscript and look forward to exploring these aspects in future work.

---

### Official Review · Reviewer_CPTS · 2024-07-12

**Soundness:** 3
**Presentation:** 3
**Contribution:** 2
**Rating:** 6
**Confidence:** 4

**Summary:**

This paper introduces a novel metho, LoQT, to address the computational and memory challenges in training large neural networks. LoQT factorize the weight matrices of the neural network into low-rank components P and B. During training, only
B is updated while P and weights are freezed. Similar to ReLoRA, the PB is merged back into the full-rank matrix W periodically. The method enables the training of models with up to 7 billion parameters on a consumer-grade 24GB GPU, and has the potential to train 13B models.

**Strengths:**

1. The combination of low-rank adaptation and quantization can significantly reduce the memory footprint.
2. This method shows end-to-end memory reduction result.
3. The introduction of quantization does not lead to significant degradation, even at 4-bit.
4. The baseline setting in the experiments is very clear and thorough.

**Weaknesses:**

1. The techniques are not very new, and can be somehow viewed as a combination of existing methods like GaLore, ReLoRA, LoftQ(compensate for quantization error by reinitialization) and NF4 quantization (from QLoRA)
2. More experiments is preferred. For example, more fine-tuning experiments on LLMs.

**Questions:**

1. Is there a specific reason why you apply NF4 in your method? Does it gives much better performance compared with FP4 (E2M1 and E3M0) and INT4?
2. Figure 1 is based on a fine-tune setting with token-number = 256. Can you provide an analysis of the memory consumption for a pretraining task, with a common batch size and sequence length setting?

---

> ### Author Rebuttal · Authors · 2024-08-07
>
> We thank the reviewer for their detailed feedback and for highlighting the strengths of our work.
>
> ## 1. More fine-tuning experiments
> Good suggestion. We have included fine-tuning experiments of a 7B model in the **common response**. We adopt a Llama model for a new low-resource language and a commonly used benchmarking task, GSM8K, demonstrating that LoQT also performs well in downstream applications.
>
> ## 2. Comparison to prior methods
> While our approach combines and builds on ideas explored in prior work like GaLore, ReLoRA, LoftQ, and NF4 quantization, we believe our work meaningfully extends these both in terms of how we deviate from the prior individual works and how we combine them.
>
> We show that for the first time, LoRA can efficiently and successfully be used for pretraining while enabling the quantization of most weights with 4-bit precision, resulting in significant memory savings. The exponentially increasing update intervals, combined with the error compensation and gradient-based initialization, all come together holistically, where the combined benefits are larger than the sum of the parts.
>
> 1. **Separation into Trainable and Frozen Weights**
>    - **GaLore:** Derives a low-rank projector using an SVD of the gradient. However, Galore does not separate trainable weights from frozen weights. In Galore, only the projector matrix $P$ is not trained, while the weight matrix $W$ is updated contiguously at every update step.
>    - **LoQT:** We extend this by introducing a separation between frozen and trainable weights. Both $W$ and one of the factors (the projector $P$) are frozen while the new factor $B$ is trained. This further reduces memory for gradients (only low rank, whereas Galore uses "full rank" gradients) and enables pre-training within the LoRA framework. This then enables quantizing most of the model weights, as discussed below.
>
> 2. **Gradient Based Initialization and Periodic Merging**
>    - **ReLoRA:** The $A$ and $B$ factors are initialized using Gaussian noise and zeros, and both are trained. ReLora then periodically merges the product of the factors into the weights of $W$ to explore a new subspace. However, ReLoRA requires full rank training for the first 1/3 of the time to be effective, limiting its usability on smaller GPUs.
>    - **LoQT:** We build on ReLoRA by using clearly separated frozen and trainable weights. We show that to attain well-performing pretraining using just low-rank adapters, a gradient-based initialization of the LoRA factors is needed. This enables quantization approaches like NF4 from QLoRA while maintaining comparable precision to regular FP16 training. We only train a single low-rank factor per layer, which enables quantizing the weights of both $W$ and $P$, allowing for additional memory savings.
>
> 3. **Quantized Pre-training**
>    - **QLoRA:** Works well for fine-tuning but not pre-training. However, implementing this with ReLoRA, without gradient-based initialization, would face similar problems, and quantization could exacerbate these issues.
>    - **LoQT:** We use the specific NF4 quantization approach from QLoRA, as it has been shown to be superior to FP4 and INT4 quantization by a significant margin. We apply their approach to $W$ and $P$ by introducing error compensation and exponentially increasing update intervals:
>        - **Error Compensation Method:** We introduce an error compensation method for initializing the $B$ matrix, incorporating the quantization error between $W$ and $Q$. This significantly reduces the performance gap between non-quantized and quantized models. We show (Figure 1 in the extra PDF) that simple quantization of $W$ and $Q$ creates a significant gap in model quality. The method reduces this gap by 72%. This is impossible with GaLore due to the continuous updates of $W$. While LoftQ also uses error compensation, their formulation operates under different constraints (see response 4 to Ekw9).
>        - **Adaptive Update Intervals:** We vary the length of update intervals by exponentially increasing intervals for selecting a new projection matrix. This, in combination with the error compensations, reduces the error by 93%. This opens up further research opportunities into how the choice of update intervals affects learning and runtime.
>
> In summary, our contributions nontrivially build on prior ideas of low-rank training and quantization, extending and combining them in ways that were previously not possible, with significant memory improvements and observed performance gains.
>
>
> ## 3. What is the reasoning for choosing NF4?
> We chose NF4 since it performs better than FP4 (E2M1 and E3M0) and INT4 in related work. As shown in Table 2 (Page 7) of QLoRA, NF4 achieves a significantly better mean perplexity (PPL) than the other quantization methods. While our focus was primarily on reducing the memory footprint of pre-training, we acknowledge that INT4 could enable faster computations since it is directly supported on certain GPUs. However, to gain this benefit, one must do low-precision matrix multiplications, which we see as an interesting future direction.
>
> ## 4. Can you provide an analysis of memory savings with varying sequence lengths?
> Good question. With larger contexts, the overall memory consumption is increasingly influenced by activations. Following prior work, our experimentation has focused on the setting of shorter context lengths (256 tokens). But as demonstrated in Figure 3 in the extra PDF, the benefit of LoQT does transfer to longer context lengths, enabling training of Llama 7B on consumer hardware with a context length of 2048 and 4096 on a 40GB A100 without activation checkpointing.

---

> > ### Comment · Reviewer_CPTS · 2024-08-08
> >
> > Thanks for your rebuttal. Your explanations are good. I decide to increase my score accordingly.

---

### Official Review · Reviewer_ps9K · 2024-07-13

**Soundness:** 3
**Presentation:** 3
**Contribution:** 3
**Rating:** 6
**Confidence:** 3

**Summary:**

LoQT is a method for low-rank training of high-rank quantized weights, using gradient-based tensor factorization to initialize low-rank trainable weight matrices. LoQT is suitable for both pretraining and fine-tuning models, significantly enhancing its applicability. Despite the 24G GPU memory limitation, LoQT can train a 13b model with only a slight speed reduction.

**Strengths:**

LoQT can significantly reduce GPU memory requirements for training LLMs with only a minor speed drop. It enables training LLMs from scratch. Gradient-based tensor factorization allows LoQT to freeze and quantize low-rank parameters, further enhancing efficiency.

**Weaknesses:**

The model size involved in the experiment is too small. Considering that LoQT has always used 24G 4090 as its efficiency target, the experiment only covers the model size that can be fully trained in 24G, which cannot highlight the advantages of LoQT.

**Questions:**

1. Given the computational power demonstrated in row 213 and the excellent efficiency of LoQT, it is surprising that the paper does not present results for larger models. What factors prevent fine-tuning on larger models? Are there any part that only use simulated results without achieving actual speedups?
2. In Figure 1, LoQT A8bit LW is shown as the most memory-efficient method. Is there any section in the paper where the results of this most extreme version are presented?
3. Table 3 states that LoRA cannot be used for pretraining. This raises confusion about the data in the LoRA row of Table 1, which focuses on pretraining methods. It appears this might be a mistake carried over from GaLore.

---

> ### Author Rebuttal · Authors · 2024-08-07
>
> We thank the reviewer for the feedback and recognition of LoQT’s memory efficiency and its practical applicability. % for both pretraining and fine-tuning through gradient-based tensor factorization.
>
> ## 1. Size of models
> We agree that training larger models than 1B would strengthen our work. In particular, the suggestion of fine-tuning on larger models. We have added experiments for fine-tuning a 7B parameter model to better showcase the potential of LoQT on larger scales. We show this successfully in the **common response**. See Tables A and B in the common response, and Figure 2 in the extra PDF.
>
> While we would have liked to train models >1B from scratch, these require computational resources beyond those that have been available to us. As reported in Sec. 4.1 (Line 214), training a 1B model takes more than four days on four A100 GPUs for Galore, Regular FP16 training, and LoQT.
> Our aim was to validate our method across different model sizes to demonstrate that LoQT can achieve the same performance as regular FP16 training while maintaining most weights in a 4-bit quantized format and performing only low-rank updates. With this validation, we show that LoQT is effective in terms of quality while using much less memory.
>
>
> ## 2. Memory Efficiency in Figure 1
> To clarify the differences between memory savings with layer-wise gradient computations and 8-bit Adam, we conduct an ablation experiment using a 130M parameter model.
> This experiment compares four settings: regular training, 8-bit Adam, layer-wise gradient updates, and a combination of 8-bit Adam with layer-wise updates, tested on Galore, LoQT, and regular FP16 training.
> Our results, illustrated in Figure 4 in the extra PDF, show that adding these memory-saving components introduces a small decrease in performance. Importantly, LoQT experiences a proportionally smaller decrease in performance compared to Galore and full training when combining both 8-bit and layer-wise updates. These results demonstrate that while memory savings come with some trade-offs, LoQT maintains good performance. In addition, due to the lower memory requirements of LoQT, we enable training of larger models without using layer-wise gradient computations and 8-bit Adam.
>
> ## 3. Clarification on LoRA for Pretraining (Tables 1 and 3)
> We acknowledge the confusion regarding Table 3’s statement that LoRA cannot be used for pretraining. To clarify, while LoRA does not perform well for pretraining, it is not entirely unusable. As demonstrated by both Galore and ReLoRA, LoRA is comparatively less effective at pretraining.
> We will update Table 3 to indicate “Yes/Sort-of” and reference ReLoRA, which shows that even with a warm start (full training for 1/3 of the iterations), the difference in perplexity between LoRA and regular pre-training is still significant.

---

> > ### Author Response · Authors · 2024-08-12
> >
> > Dear reviewer, thank you for taking the time to consider our rebuttal. We look forward to hearing if you find it answers your concerns.

---

> > > ### Comment · Reviewer_ps9K · 2024-08-13
> > > **Response to rebuttal**
> > >
> > > Thank you for the clarification. I will maintain my score.

---

### Official Review · Reviewer_tC4j · 2024-07-22

**Soundness:** 3
**Presentation:** 3
**Contribution:** 3
**Rating:** 6
**Confidence:** 3

**Summary:**

The authors propose LoQT, a novel method to use low-rank LoRA adaptors directly during pretraining. LoRA adaptors have been used for downstream finetuning before, but not for full pretraining.

LoQT works by first quantization a weight matrix $W$ into $W_q$. Then the quantization error is decomposed into low-rank factors. One factor is the projection $P$ from the low-rank subspace spanned by the principle eigenvectors of the current weight gradient (obtained via SVD on the gradient ), and the other is a trainable factor $B$. $P$ is frozen and also quantized. LoQT will repeat the above initialization process several times over training, and essentially represents the final weight as a sum of low-rank weight updates, each initialized via SVD of the weight update on a certain step.

LoQT leverages previous observation that the weight gradient is low-rank, and that this rank decreases over training as learning rate decreases. The initialization frequency is set of increase over training.

Experiments show that LoQT improves greatly over LoRA variants for pre-training, and improves in downstream accuracy over GaLore, a previous method which also tackles using LoRA adaptors for pretraining.

**Strengths:**

- Interesting idea backed by empirical observation of low-rank weight gradients.
- Well written paper, good explanation of complicated idea.

**Weaknesses:**

- This type of approach should work with any type of DNN, but the authors only experiment with LLMs, and just one family of LLM (Llama-2). Some data on how generalizable LoQT is to other architectures would be welcome.

**Questions:**

None

---

> ### Author Rebuttal · Authors · 2024-08-07
>
> ## Generalization to other architectures and models
>
> We agree that LoQT should work with any type of DNN using linear layers, such as vision transformers or state space models. To narrow the scope of our work and provide a more detailed analysis, however, we choose to focus on a well-studied auto-regressive language model and a bi-directional masked language model that have been commonly used as a basis in much of the related work. We believe that Llama 2 is representative of a large family of text-based Transformer LLMs, which overlap in most of the underlying architectural decisions. In Sec. 4.2 (line 231) and Tab. 2, we present results for a diverse set of model sizes ranging from 60M to 1B parameters within the Llama 2 architecture. In Sec. 5 (line 276) and Fig. 4, we provide an ablation study of each model component, as well as the effects of varying update interval sizes. In Sec. 4.3 (line 241) and Tab. 2, we show results for fine-tuning a bi-directional language model, the DeBERTaV3-base model on the GLUE tasks. All experiments are averaged over three seeds to ensure robustness.
>
> Adding to this, we include more fine-tuning experiments of a pre-trained model in the **common response**. These include continued pre-training of the Llama2 7B model on a low-resource language and adaptation on a common benchmarking task (GSM8K) that confirms the memory saving and performance preservation using LoQT for adaptation of an existing model.
>
> Having said that, we do plan to consider LoQT's generalizability in future work and thank the reviewer for pointing out this strength of LoQT. We will make sure to elaborate on this in the updated manuscript, including what needs to be kept in mind when using quantization for, e.g., vision models.

---

> > ### Author Response · Authors · 2024-08-12
> >
> > Dear reviewer, thank you for taking the time to consider our rebuttal. We look forward to hearing if you find it answers your concerns.

---

### Author Rebuttal · Authors · 2024-08-07

We appreciate the constructive and positive reviews of our work. All reviewers point out the significance of the memory reduction of LoQT in pre-training, and that our experiments show that "quantization does not lead to significant degradation" **CPTS** while "significantly reducing GPU memory requirements for training LLMs" **ps9K**. **tC4j** and **CPTS** find the work "novel". **tC4j** and **ps9K** acknowledge that adapters have not been used for full pretraining before. **Ekw9** points out the benefit of the "new periodical update strategy". **tC4j** finds the paper "well written" and having a "good explanation of a complicated idea" while **CPTS** finds the setting of the experiments "very clear and thorough".

A common theme in the reviews is suggesting experimentation with larger model sizes than 1B, such as fine-tuning a publicly available pre-trained LLM. We agree that these are valuable experiments and include two kinds of fine-tuning results for Llama2-7b. The first is a task-adaption on grade school math questions using the GSM8K dataset [2]. The second is continued pre-training for a low-resource language. The results show that LoQT gives excellent results on both tasks while also reaping the benefits of the significantly lower memory footprint established for the pre-training tasks. This is described in detail below.

Reviewers **CPTS** and **Ekw9** comment on how LoQT compares to prior work. To address these, we describe in detail the key differences to prior work in the individual responses, as well as how we non-trivially combine them to make LoQT work.

We respond to other comments in the individual responses, we encourage all reviewers to have a look at them. Finally, we would like to thank all reviewers for the quality of their reviews.

## Task adaptation for GSM8K on Llama 7B
The GSM8K dataset [2] "is a dataset of 8.5K high-quality linguistically diverse grade school math word problems created by human problem writers. The dataset is segmented into 7.5K training problems and 1K test problems". It has been used extensively to benchmark LLMs and is a good candidate for evaluating LoQT. We experimented with models of sizes 7B and 13B, as suggested by the reviewer. We adopt the evaluation and training setup of ApiQ with a sequence length of 512 for 4-bit quantization and finetuning of Llama-2 models (7B and 13B) on the GSM8K training set [2, 3] and report the accuracy on the test set. Using the same hyperparameters as reported by ApiQ (see Tables A4 and A5 of their paper), LoQT achieves an accuracy of 41.6 for Llama-7B and 52.2 for Llama-2 13B. In contrast, the previous best were 39.8 and 51.2, thus surpassing the baselines. Both results are without merging and with rank 64. Results are shown in the Table A below.


| Method | Bit | GSM8K (acc↑) Llama-2-7B | GSM8K (acc↑) Llama-2-13B |
|--------|-----|--------------------------|---------------------------|
| LoRA   | 16  | 36.9                     | 45.3                      |
| QLoRA  | 4   | 35.1                     | 39.9                      |
| LoftQ  | 4   | 35.0                     | 45.0                      |
| ApiQ   | 4   | 39.8                     | 51.2                      |
| LoQT   | 4   | **41.6**                 | **52.2**                  |

*Table A: Finetuning results of GSM8K on Llama-2-7B and Llama-2-13B.*

## Continued pre-training of Llama 7B
We run our pre-training experiments for models up to 1B parameters. To demonstrate the feasibility of LoQT on a larger model size in a practical application, we do continued pre-training for language adaptation. To this end, we start with the `meta-llama\Llama-2-7b-hf` from Hugging face and run 1500 updates (1 epoch) on the model using a batch size of 512. We compare LoQT with NF4 quantization, LoQT-nq (no quantization), regular training and GaLore. We use rank 1024 for all models where applicable, adam8bit optimizer, 150 warmup steps. For LoQT and GaLore we use a learning rate of 0.0002 with a scale factor of 0.25 and for the others a learning rate of 0.00005. The dataset we train on is a curated subset of a public Icelandic text dataset [1] with ~770k documents, and a corresponding evaluation set, we will release the splits when appropriate. We choose Icelandic since the model has a limited performance on the language yet it was included to some degree in the pre-training data, enabling a clear improvement trajectory.
Validation perplexity results are shown in the Table B below and as a step vs perplexity plot in Figure 2 in the extra PDF.
LoQT and LoQT-nq perform the best at 3.61 and 3.63 in perplexity, similar to full training at 3.79, while GaLore gets 3.96 and the original model 4.90.

| Model          |  Perplexity ↓ |
|----------------|------------------|
| No Training    | 4.90           |
| Full (FP16)    | 3.79           |
| GaLore         | 3.96           |
| LoQT-nq        | **3.61**           |
| LoQT           | 3.63           |

*Table B: Validation perplexity continued pre-training of Llama 7B on Icelandic corpus.*

We also include additional experiments: 1) memory component ablation experiments (Figure 4 in the extra PDF), showing the effect of using Adam 8bit and layer-wise gradients; 2) experiments demonstrating LoQT’s memory scalability for training Llama 7B with longer context lengths up to 2048 on consumer hardware (Figure 3 in the extra PDF); and 3) an experiment showing fine-tuning results for GLUE without merging factors into $W$ during training (Table 1 in the extra PDF).

## New References

[1] Barkarson, S., et. al., (2022, June). Evolving Large Text Corpora: Four Versions of the Icelandic Gigaword Corpus.

[2] Cobbe, K., et. al., (2021). Training Verifiers to Solve Math Word Problems.

[3] Liao, B., et. al., (2024). ApiQ: Finetuning of 2-Bit Quantized Large Language Model

---

### Decision · Program_Chairs · 2024-09-25

**Decision:**

Accept (poster)

**Comment:**

Reviewers agree that the paper represents a novel contribution in efficient pretraining, albeit being a combination of existing ideas.

Some questions about the scale of the experiments are well answered in the rebuttal, and reviewers have concurred that their questions are well answered.